# From stable Sb- and Bi-centered radicals to a compound with a Ga=Sb double bond

Chelladurai Ganesamoorthy[1], Christoph Helling[1], Christoph Wölper[1], Walter Frank[2], Eckhard Bill[3], George E. Cutsail III[3] & Stephan Schulz [1]

Neutral stibinyl and bismuthinyl radicals are typically short-lived, reactive species. Here we show the synthesis and solid-state structures of two stable stibinyl [L(Cl)Ga]$_2$Sb· **1** and bismuthinyl radicals [L(I)Ga]$_2$Bi· **4**, which are stabilized by electropositive metal centers. Their description as predominantly metal-centered radicals is consistent with the results of NMR, EPR, SQUID, and DFT studies. The Lewis-acidic character of the Ga ligands allow for significant electron delocalization of the Sb- and Bi- unpaired radical onto the ligand. Single-electron reduction of [L(Cl)Ga]$_2$Sb· gave LGaSbGa(Cl)L **5**, the first compound containing a Ga=Sb double bond. The π-bonding contribution is estimated to 9.56 kcal mol$^{-1}$ by NMR spectroscopy. The bonding situation and electronic structure is analyzed by quantum mechanical computations, revealing significant π backdonation from the Sb to the Ga atom. The formation of **5** illustrates the high-synthetic potential of **1** for the formation of new compounds with unusual electronic structures.

[1] Institute for Inorganic Chemistry and Center for Nanointegration Duisburg-Essen (Cenide), University of Duisburg-Essen, Universitätsstraße 5-7, D-45117 Essen, Germany. [2] Institute for Inorganic Chemistry and Structural Chemistry, Heinrich Heine University Düsseldorf, Universitätsstraße 1, D-40225 Düsseldorf, Germany. [3] Max-Planck-Institute for Chemical Energy Conversion (CEC), Stiftstrasse 34-36, D-45470 Mülheim an der Ruhr, Germany. Correspondence and requests for materials should be addressed to G.E.C . (email: george.cutsail@cec.mpg.de) or to S.S. (email: stephan.schulz@uni-due.de)

F irst-row main group radicals are known since Gomberg reported on the first persistent triphenylmethyl radical[1], whereas radicals of heavier main group elements[2–4], which represent important intermediates in chemical and material synthesis[5,6], are less explored. The development of new strategies, e.g., kinetic stabilization by use of bulky substituents and electronic stabilization of the singly occupied molecular orbital (SOMO) by placing the radical center adjacent to an electronegative atom, gave access to persistent and stable radicals. Moreover, radicals were stabilized by $\sigma$-donating carbenes, e.g., cyclic alkyl(amino)carbenes (cAAC)[7–11].

In contrast to well established P-centered radicals[12–23], only a few Sb and Bi-centered radicals were reported (Fig. 1)[24]. Laser-ablated Sb and Bi atoms react with hydrogen and methane to give short-lived Sb(II) and Bi(II) radicals $H_2E\cdot$ and $HMeE\cdot$ (E = Sb, Bi)[25,26] and persistent cAAC-stabilized stibinyl (a)[27] and bismuthinyl radicals (b) were synthesized in solution[27,28]. c is the only structurally characterized neutral bismuthinyl radical[29], while d and e represent the only stable anionic and cationic Sb-centered radicals[30,31].

Continuing our investigations into Sb-X (X = Sb, C, N, Cl) bond activation reactions with monovalent Mg(I) and Ga(I) compounds[32–36], here we report on the synthesis and characterization of stable neutral stibinyl and bismuthinyl radicals [L (X)Ga]$_2$E· (E = Sb, X = Cl 1, E = Bi, X = I 4) and the first compound containing a Ga=Sb double bond, LGaSbGa(Cl)L 5.

## Results

**Syntheses**. The reaction of two equivalents of LGa {L = HC[C (Me)N(2,6-$i$Pr$_2$C$_6$H$_3$)]$_2$} with Cp*SbCl$_2$ (Cp* = C$_5$Me$_5$) gave stibinyl radical [L(Cl)Ga]$_2$Sb· 1 (Fig. 2), traces of L(Cl)GaSb(H)Cp* 2, LGaCl$_2$ and distibene [L(Cl)Ga]$_2$Sb$_2$[36], whereas the equimolar reaction yielded L(Cl)GaSb(Cl)Cp* 3, which decomposed to LGaCl$_2$, decamethylfulvalen (Cp*$_2$) and antimony metal (Supplementary Figs. 11, 12). The formation of 1 upon reaction of 3 with a second equivalent of LGa is faster than the decomposition of 3 as was shown in an independent experiment with isolated 3. Analogous reactions of LGa with Cp*BiI$_2$ yielded bismuthinyl radical [L(I)Ga]$_2$Bi· 4, whereas the equimolar reaction only gave LGaI$_2$, Cp*$_2$ and Bi metal even at −30 °C, pointing to an expressed temperature lability of the monoinsertion product L(I)GaBi(I) Cp*.

The stepwise insertion of LGa into the E-X bonds of Cp*EX$_2$ followed by homolytic bond cleavage of the E-Cp* bond and elimination of Cp*$_2$ are the key steps in the formation of 1 and 4. Their formation illustrates the high synthetic potential of LGa for the synthesis and stabilization of unusual electronic structures. In addition, it demonstrates for the first time the stabilizing effect of the coordination of an electropositive metal at the radical center. The phosphorus radical [·P{NV[N(Np)Ar']$_3$}$_2$] (Np = Neopentyl, Ar' = 3,5-Me$_2$C$_6$H$_3$), which is resonance stabilized by a vanadium (IV/V) redox couple, represents the only metal-stabilized radical, but misses a direct bond of the metal to the radical center[37].

**Solid-state structures**. (Supplementary Table 1) 1 (Fig. 3a) and 4 (Fig. 3b) adopt V-shaped geometries with Ga-E-Ga bond angles of 104.89(1)° (1) and 106.68(3)° (4), while the bismuthinyl radical c shows a significantly smaller N1-Bi-N2 bond angle (94.70(16)°), resulting from embedding the Bi radical center into the ring system. P- and As-centered radicals ($^{Dipp}$NHC)PEP($^{Dipp}$NHC) (E = P 90.53(3)°; As 86.53(4)°; Dipp = 2,6-$i$Pr$_2$C$_6$H$_3$)[20], in which the single-electron occupies a $\pi^*$-orbital, showed smaller P-E-P bond angles due to the less bulky ($^{Dipp}$NHC)P groups. 1 and 4 show *syn,syn* configurations with the halogen atoms orienting toward the middle of the Ga-E-Ga skeleton. The Ga–Sb bond

lengths in 1 [2.5899(4), 2.5909(3) Å] are comparable to those in [L(X)Ga]$_2$Sb$_4$ [X = Cl 2.6008(13) Å, NMe$_2$ 2.5975(5) Å][33,36], but shorter than in 2 [2.6265(11) Å], 3 [2.6979(2) Å], Ga-substituted distibenes [L(X)Ga]$_2$Sb$_2$ [X = NMe$_2$ 2.6477(3) Å, NMeEt 2.6433 (6) Å, Cl 2.6461(2) Å][33,36], the sum of the covalent radii (Ga 1.24 Å; Sb 1.40 Å)[38], as well as Lewis acid-base adducts R$_3$Ga-SbR'$_3$ (2.84–3.02 Å) and heterocycles [R$_2$GaSb(SiMe$_3$)$_2$]$_x$ (x = 2, 3; 2.65–2.76 Å)[39]. The Ga–Bi bond lengths in 4 [2.6640(9), 2.6663 (9) Å] agree well with those of [L(OSO$_2$CF$_3$)Ga]$_2$Bi$_2$ (2.655(1) Å)[40], but are shorter than those in [L(OC$_6$F$_5$)Ga]$_2$Bi$_2$ [2.693(6) Å][40], L(Et)GaBiEt$_2$ [2.6959(3) Å][41], LGa(BiEt$_2$)$_2$ [2.6961(6) Å, 2.7303 (10) Å][32] and the sum of the covalent radii (Ga 1.24 Å; Bi 1.51 Å)[38]. The Cp* rings in 2 (Supplementary Fig. 30) and 3 (Supplementary Fig. 31) adopt $\eta^1$ coordination modes (Sb-C 2.253(9) Å 2; 2.2381(16) Å 3) and the presence of the Sb-H moiety in 2 was discerned in the electron density map.

**Solution-phase characterization**. 1 can be stored without decomposition under argon atmosphere for months while 2–4 have to be cooled to −35 °C (Supplementary Figs. 7, 11). The $^1$H and $^{13}$C NMR spectra of 2 (Supplementary Figs. 4, 5) and 3 (Supplementary Figs. 8, 9) show resonances of the $\beta$-diketiminate ligand and Cp* moieties and the presence of the Sb-H group in 2 was confirmed by $^1$H NMR (singlet at 2.73 p.p.m.) and IR spectroscopy (Supplementary Fig. 6, $\nu_{Sb-H}$ 1855 cm$^{-1}$). The $^1$H NMR spectra of 1 and 4 (Supplementary Figs. 1, 13) in toluene-$d_8$ show broad signals as is typical for paramagnetic species. The resonances shift to lower field upon cooling (Supplementary Figs. 2, 14), whereas no indications of dimerization into the respective diamagnetic distibine and dibismuthine was found even at −80 °C. The paramagnetic character in solution was further confirmed by determining the magnetic susceptibility of 1 (1.68 $\mu_B$) and 4 (1.61 $\mu_B$) by use of the Evans method[42], which is consistent with the presence of a single-unpaired electron, and by electron paramagnetic resonance (EPR) spectroscopy.

**EPR Spectroscopy**. Compound 1 in both the solid and frozen solution exhibits a complex continuous-wave electron paramagnetic resonance spectrum at X-band (~9.6 GHz) frequency from the S = 1/2 antimony radical center (Supplementary Fig. 27). The solid and frozen solution of 1 exhibit equivalent EPR spectra, however, the latter has narrower linewidths and is more ideal for high-quality simulations and further analysis. The EPR spectrum of the frozen solution of 1 exhibits a complex super-hyperfine pattern from the mixtures of the antimony and gallium natural abundance nuclear active isotopes. To accurately determine g and the principle A-values, the EPR spectrum was also collected at Q-band (34 GHz), Fig. 4. The high-field edges of both the X- and Q-band spectra of 1 exhibit a resolved hyperfine fine pattern of the $g_3$ feature belonging to the $|M_I = +5/2>$ hyperfine transition of the $^{121}$Sb isotopologue. This single feature is further split by two equivalent Ga atoms to yield a 5 line pattern. This allows for the accurate determination of the $g_3 = 1.967$, and $A_3$ values for the Sb atom ($A_3(^{121}$Sb) = 1138 MHz) and the two Ga atoms ($A_3(^{69}$Ga) = 134 MHz) by simulation. The remaining EPR parameters were elucidated by simultaneous fitting of the X- and Q-band spectra and are reported full within the caption of Fig. 4.

Evaluation and decomposition of the hyperfine tensor and its isotropic and dipolar components (further discussed in the Supplementary Methods) yields a small $a_{iso}(^{121}$Sb) = 86 MHz value that indicates negligible unpaired s orbital population at the Sb atom ($\rho$(Sb s) < 0.003). The dipolar tensor yields insight into the axial unpaired spin population, i.e., the p-orbital character. For the determined value of t = 526 MHz, the unpaired p-orbital spin population of $\rho$(Sb p) = 0.837. The two coordinated Ga

**Fig. 1** Sb- and Bi-centered radicals. Persistent radicals (**a**, **b**), which have been identified in solution by electron paramagnetic resonance (EPR), and stable radicals of the heaviest group 15 elements (**c**-**e**), whose solid-state structures were determined by single-crystal X-ray diffraction

**Fig. 2** Synthesis of **1**–**4**. **i** stirring LGa and 0.5 eq. of Cp*SbCl$_2$ at 25 °C in benzene for 7 d with elimination of 0.5 Cp*$_2$. **ii** stirring LGa and 0.5 eq. of Cp*BiI$_2$ from −40 to 25 °C in toluene over 4 h with elimination of 0.5 Cp*$_2$. **iii** stirring LGa and an equimolar amount of Cp*SbCl$_2$ in benzene at 25 °C for 1 h. **iv** stirring **3** in benzene at 25 °C for 6 d

atoms exhibit equivalent hyperfine couplings of $A(^{69}Ga)$ = [132, 176, 134] MHz, $a_{iso}$ = 147 MHz with $\rho(Ga\ s)$ = 0.012. Given the p-orbital character of the SOMO, a significant dipolar character is anticipated for the Ga atoms also. The resultant $t \sim 14.3$ MHz value estimates an Ga p-orbital unpaired spin population of $\rho(Ga\ p)$ = 0.066. The sum of these spin population estimates is 0.994 = $\rho(Sb\ p)$ + 2x[$\rho(Ga\ s)$ + $\rho(Ga\ p)$], accounting for nearly all of the unpaired spin of the radical. The Ga unpaired spin population, $\rho(Ga)$, of **1** determined by the Ga hyperfine above is significantly larger than other ligands of heavy group 15 radicals[6,29], owing most likely to its unique metallic character.

The CW X-band EPR of **4** exhibits a complex spectrum (Supplementary Fig. 28) of hyperfine transitions from 0 to ~7000 G resultant from the extremely large intrinsic hyperfine couplings ($a_0$ = 77,530 MHz, $b_0$ = 664 MHz) of the 100% natural abundance, $I$ = 9/2, $^{209}$Bi isotope[43]. The spectrum is further complicated by

additional hyperfine splitting of the two equivalent Ga atoms, as previously observed in **1**. Attempts to refine the EPR parameters of **4** through Q-band EPR were moderately successful. Limits upon the g-values and Bi hyperfine may be placed, including $g_{av} \sim$ 2, but satisfactory simulation of the spectrum has not been successful. The paramagnetic spectrum observed is undoubtedly Bi centered, localized in a p-orbital. There exists minimal Bi s orbital spin population, otherwise, the observed hyperfine would be (1) significantly larger than observed, or (2) not measurable at both X and Q-band frequencies. SQUID measurements of a solid sample of **4** were performed to prove its radical character. A temperature-independent effective magnetic moment of 1.78 $\mu_B$, close to the spin-only value for $S$ = 1/2, was found in the range of 10–300 K, as was expected for a magnetically isolated radical compound (Supplementary Fig. 29). A simulation yielded $g_{av}$ = 2.05, which nicely confirms that (**4**) is a pure radical compound.

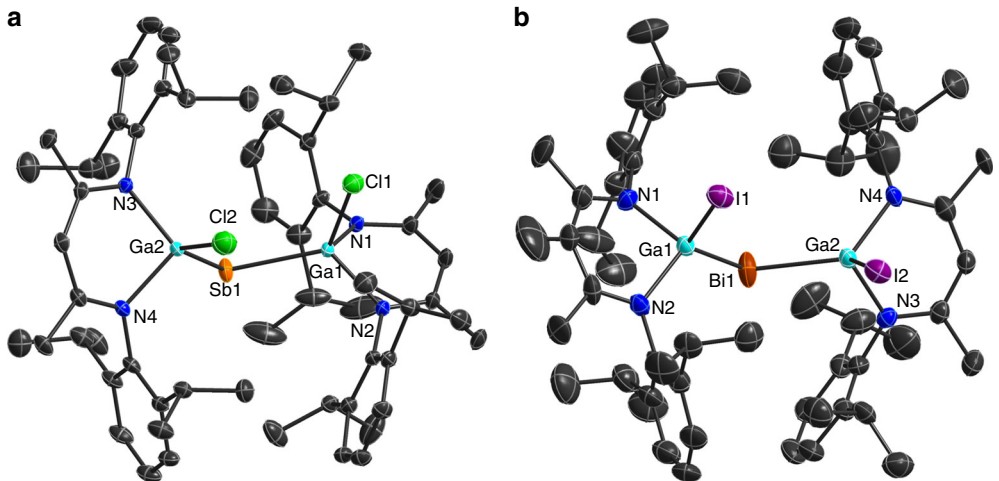

**Fig. 3** Molecular structures of **1** and **4**. Molecular structures of **1** (**a**) and **4** (**b**) in the solids of **1** and **4** · 2C$_7$H$_8$. H-atoms were omitted for clarity, displacement ellipsoids are drawn at the 50% probability level. Selected bond lengths and angles: **1**, Ga–Sb 2.5899(4), 2.5909(3) Å, Ga–Sb–Ga 104.89(1)°; **4**, Ga–Bi 2.6640(9), 2.6663(9) Å, Ga–Bi–Ga 106.68(3)°

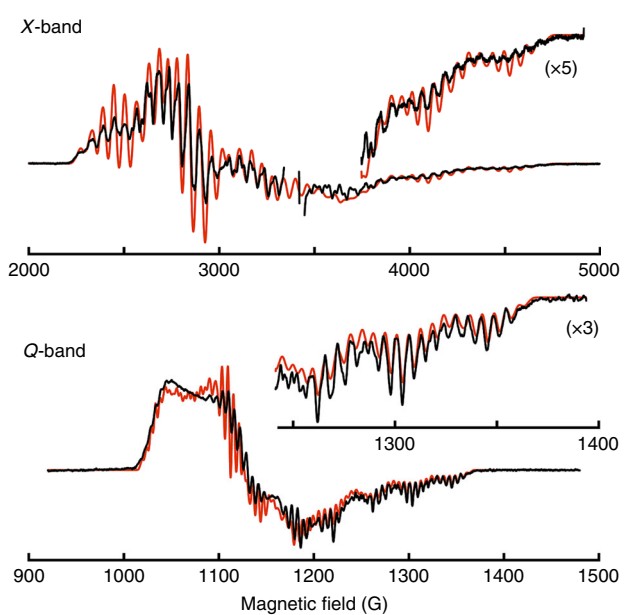

**Fig. 4** EPR spectroscopy. Continuous-wave EPR spectra of **1**. as frozen solution obtained at *X*-band (9.634 GHz) and *Q*-band (34.202 GHz) in black with simulated spectra in red. High-field regions of each spectra are expanded as insets. **g** = [$g_1$, $g_2$, $g_3$] = [2.298 2.114 1.967], **A**($^{121}$Sb) = [$A_1$, $A_2$, $A_3$] = [−385, −496, 1138] MHz and 2 × **A**($^{69}$Ga) = [132, 176, 134] MHz, lw (peak–peak) = 2.0 G. *X*-band conditions: temperature 15 K; modulation amplitude = 6 G; modulation frequency = 100 kHz; time constant = 20.48 ms; scan time = 167 s; single scan. *Q*-band conditions: temperature 10 K; modulation amplitude = 6 G; modulation frequency = 100 kHz; time constant = 14.65 ms; scan time = 60 s; number of scans = 24

**Quantum chemical calculations.** The assignments of **1** and **4** as Sb and Bi p-orbital centered radicals is supported by quantum mechanical computations. DFT geometry optimized structures of **1′** and **4′** (Supplementary Tables 2, 3) resembles those found by crystallography, with slightly elongated Ga-E bond length (E = Sb 2.655, 2.658 Å **1′**; Bi 2.748, 2.743 Å **4′**) and wider Ga-E-Ga bond angles (E = Sb 111.05° **1′**; Bi 113.4° **4′**). The SOMOs are localized Sb and Bi p-orbitals and Löwdin spin population analyses reveal 0.80 Sb + 0.16 Ga (2 × 0.08 Ga) spin densities for **1′** and 0.81 Bi +

0.15 Ga (2 × 0.074 Ga) for **4′**, in excellent agreement with the EPR analysis of **1** (Fig. 5). Natural bond orbital (NBO) analyses of the Ga-E bonds (1.91e occupancy **1′**; 1.90e occupancy **4′**) show similar Ga-E bond polarization, with slightly more electron population on the Sb (~62% pop; 11.0(2)% s, 88.1(2)% p) and Bi atoms (59.1% pop; 7.6% s, 92.1% p) compared to the sp hybridized Ga atoms (~38% pop; 44.1(7)% s, 55.6(0)% p **1′**; 40.9% pop, 45.4% s, 54.2% p **4′**). The Sb and Bi centers each possess a fully occupied lone pair orbital of predominantly s character (1.91e occupancy, 77.5% s, 22.4% p **1′**; 1.90e occupancy, 84.8% s, 15.2% p **4′**) and the SOMO of full p character (0.93e occupancy, 0.15% s, 99.8% p, 0.19% d **1′**; 0.93e occupancy, 0.06% s, 99.8% p, 0.14% d **4′**). The Lewis-acidity of the Ga ligands allow for significant electron delocalization of the Sb and Bi unpaired radical onto the ligand.

**Reactivity Studies.** **1** is stable in toluene-$d_8$ solution up to 90 °C and shows no sign of decomposition within 15 h, whereas it slowly decomposes at 130 °C within 5 days with formation of LH and LGa, while metal formation was not observed. In contrast, **4** is far more temperature labile and slowly decomposes at ambient temperature with formation of LGaI$_2$, LGa and bismuth metal. To investigate the redox properties of **1**, its reaction with the single-electron oxidant [NO][BF$_4$] was studied. The reaction proceeded with elimination of NO, fluorine abstraction[21,44] and formation of LGaClF (Supplementary Fig. 32), traces of LGaX$_2$ (X = F, Cl, Supplementary Fig. 16) and elemental Sb (Fig. 6). In contrast, the equimolar reduction reaction with the single-electron reductant KC$_8$ occurred with clean formation of LGaSbGa(Cl)L **5** (Supplementary Fig. 21), while addition of another equivalent of KC$_8$ yielded elemental antimony and LGa (Supplementary Figs. 22, 23).

**5** is stable as isolated crystals under argon atmosphere and in toluene-$d_8$ solution at ambient temperature, whereas it slowly starts to decompose in solution at 90 °C. Thermolysis at 120 °C for 30 h yielded LGa and distibene [L(Cl)Ga]$_2$Sb$_2$, which further underwent LGaCl$_2$ and LGa elimination and finally formed [L(Cl) Ga]$_2$Sb$_4$ (Supplementary Figs. 24–26)[33]. $^1$H and $^{13}$C NMR spectra of **5** show the expected resonances of the organic substituents (Supplementary Figs. 17, 19). A variable temperature $^1$H NMR study (Supplementary Fig. 18) was performed to examine the strength of the Ga–Sb π-bonding. The γ-CH signal splits into two peaks at −80 °C with a maximum peak separation of 26.7 Hz,

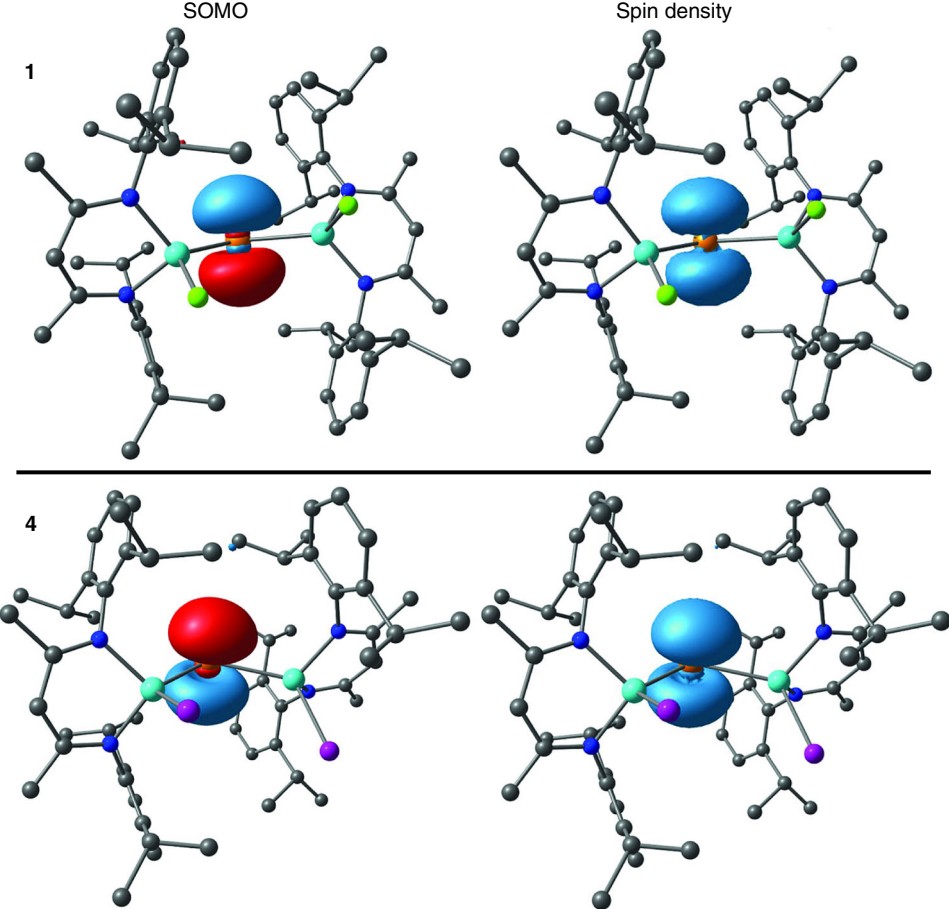

**Fig. 5** Geometry optimized structure of **1**′ and **4**′. The SOMO orbitals and spin densities exhibit the p-orbital character of the radical's unpaired electron

giving $\Delta G^{\ddagger} = 9.56$ kcal mol$^{-1}$ for the rotation about the Ga=Sb bond[45]. Power et al. reported on a π-bonding contribution in PhP(GaTrip$_2$)$_2$ (Trip = 2,4,6-$i$Pr$_3$C$_6$H$_2$) of ca. 10.2 kcal mol$^{-1}$ and addressed the restricted rotation to an allylic type of Ga–P–Ga bonding[46].

The Ga1-Sb1-Ga2 bond angle (113.184(7)°) in **5** (Fig. 7a) is significantly enlarged and the Ga1–Sb1 single bond (2.5528(2) Å) is slightly shorter compared to the corresponding values in **1** (104.89(1)°; 2.58994(4), 2.5909(3) Å). The Ga2–Sb1 bond (2.4629(2) Å) is about 9 pm shorter than in **1** and 4 pm below the sum of the calculated covalent double-bond radii (Ga 1.17 Å, Sb 1.33 Å), clearly proving the double-bond character[47]. The formation of **5** is surprising since compounds containing a double bond between heavier group 13 (Al, Ga, In, Tl) and group 15 elements (P, As, Sb, Bi) are almost unknown[48], in contrast to the large number of multiple-bonded compounds containing either boron and/or nitrogen atoms[49]. [{Li(thf)$_3$}$_2$Ga$_2${As(Si$i$Pr$_3$)}$_4$] containing a Ga=As double bond represents the only structurally characterized compound of this type, to date[50].

The DFT optimized structure of **5**′ (Supplementary Table 4) shows a Sb–Ga single-bond length of 2.639 Å (Fig. 7b), which is almost 10 pm longer than its Ga=Sb double bond (2.535 Å). The Ga–Sb–Ga bond angle in **5**′ (118.94°) is also wider than in **1**′ (111.05°), in agreement with the experimentally observed differences. The HOMO of **5**′ exhibits significant π-backdonation from the Sb to the Ga atom. NBO analysis of the Sb–Ga(L) double bond consists of a σ bond (1.90e occupancy) of significant p character (86.0% Sb, 99.6% p; 14.0% Ga(L), 99.3% p) and a π Sb–Ga(L) bond (1.85e occupancy). The π bond of **5**′ exhibits less Sb localization 55.1% (7.3% s, 92.2% p), with the

44.9% localized Ga(L) (60.1% s, 39.3% p) revealing significant π-backdonation to the approximately sp hybridized Ga atom. This bonding description contrasts the single bond (σ) between Sb–Ga (Cl,L) (1.90e occupancy) of slight Sb polarization 59.6% (10.1% s, 89.3% p) of the bond with the 40.4% Ga(Cl,L) atom (49.3% s, 50.2% p) which exhibits a typical sp Ga bonding orbital. Lastly, **5** possess only a single sp Sb lone pair (1.84e, 82.5% s, 17.4% p) compared to one lone pair orbital and the SOMO of **1**, described above.

In summary, the synthesis and full characterization including single crystal X-ray diffraction of two stable stibinyl (**1**) and bismuthinyl radicals (**4**) is reported. Their description as predominantly metal-centered radicals is consistent with the results of NMR, EPR, SQUID, and DFT studies. Single-electron reduction reaction resulted in the isolation of the first structurally characterized compound containing a Ga=Sb double bond (**5**). The π-bonding contribution was estimated to 9.56 kcal mol$^{-1}$ by NMR spectroscopy and quantum mechanical computations clearly proved significant π-backdonation from the Sb to the Ga atom.

## Methods

**General methods**. Manipulations were carried out under a dry, oxygen-free argon atmosphere, with reagents dissolved or suspended in aprotic solvents, and combined or isolated using cannula and glove-box techniques. **1**–**4** were synthesized via the reactions of LGa with Cp*SbCl$_2$ or Cp*BiI$_2$ in different molar ratios and **5** by equimolar reaction of **1** with KC$_8$, respectively, in toluene solution and isolated by low-temperature crystallization. **1**–**5** were characterized by elemental analysis, multinuclear NMR (diamagnetic compounds **2**, **3**, **5**) and EPR spectroscopy (paramagnetic compounds **1**, **4**). X- and Q-band field-sweep EPR spectra were measured both as solids under a N$_2$ atmosphere and as frozen solution in hexane (**1**) and toluene (**4**). Magnetic moments for **1** and **4** were determined using the

**1e⁻ Oxidation**

**1e⁻ Reduction**

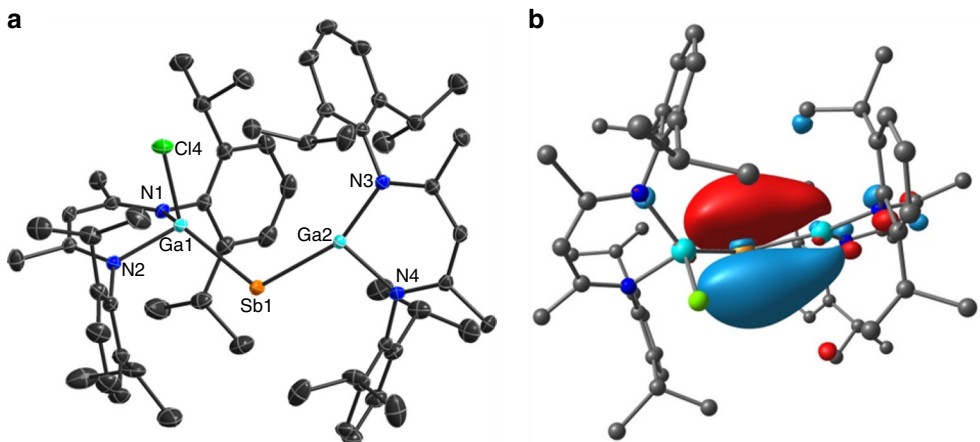

**Fig. 6** Single-electron oxidation and reduction reactions of **1**. **i** sonicating **1** and 1 eq. of NO[BF₄] in benzene at 25 °C, followed (**ii**) by addition of a 2nd eq. of NO[BF₄] in benzene at 25 °C; **iii** reaction of **1** with 1 eq. of KC₈ in benzene at 25 °C over 1 h occurred with (**iv**) elimination of KCl

**Fig. 7** Molecular and geometry optimized structures of **5** and **5′**. (**a**) Molecular structure of **5**. H-atoms were omitted for clarity, displacement ellipsoids are drawn at the 50% probability level. **b** Geometry optimized structure of **5′**, with the HOMO plotted. Selected bond lengths and angles: **5**, Ga–Sb 2.5528(2), 2.4629(2) Å, Ga–Sb–Ga 113.184(7)°; **5′**, Ga–Sb 2.639, 2.535 Å, Ga–Sb–Ga 118.94°

Evans method (solution phase) and by using a SQUID susceptometer (**4**, solid state) with a field of 1.0 T in the temperature range 2–300 K. DFT, as implemented in the ab initio quantum chemistry program package ORCA, was employed to calculate molecular geometries, orbital energies and compositions. Additional information to the synthetic, spectroscopic, crystallographic, and computational methods are given in Supplementary Methods.

**Synthesis of [L(Cl)Ga]₂Sb· 1 and L(Cl)GaSb(H)Cp* 2.** A mixture of LGa (0.297 g, 0.610 mmol) and Cp*SbCl₂ (0.1 g, 0.305 mmol) in 2 mL of benzene was stirred at room temperature for 7 days. The solvents were removed under reduced pressure and the residue was dissolved in 2 mL of hexane. The dark-red solution was kept at room temperature for 3 days to afford dark-red and tiny-yellow crystals. The red crystals were hand-picked under the microscope and fractionally crystallized in 2 mL of hexane. Fractional crystallization was repeated two times to get analytically pure form of **1**. Yield: 181 mg (0.155 mmol, 51%). M.p. 241 °C. Anal. Calcd. for C₅₈H₈₂N₄Cl₂Ga₂Sb: C, 59.67; H, 7.08; N, 4.80. Found: C, 59.55; H, 7.09; N, 4.89%. IR (neat): ν 2957, 2926, 2865, 1521, 1436, 1380, 1313, 1258, 1178, 1098, 1019, 939, 865, 798, 755, 706, 638, 614, 528, 448 cm⁻¹.

Compound **2** was isolated as colorless crystals (20 mg) while crystallizing the above reaction mixture after stirring at room temperature for 1 day in 2 mL of toluene at 8 °C. M.p. 135 °C (dec.). Anal. Calcd. for C₃₉H₅₇N₂ClGaSb: C, 59.99; H, 7.36; N, 3.59. Found: C, 60.20; H, 7.54; N, 3.54%. IR (neat): ν 2966, 2919, 2866, 1855, 1552, 1522, 1441, 1382, 1312, 1248, 1184, 1102, 1020, 933, 862, 798, 763, 635, 524, 495, 436 cm⁻¹. ¹H NMR (C₆D₆, 300.1 MHz): δ 7.23–7.02 (m, 6 H, C₆H₃(iPr)₂), 4.95 (s, 1 H, γ-CH), 3.92 (sept, ³J$_{HH}$ = 6.6 Hz, 1 H, –CH(CH₃)₂), 3.81 (sept, ³J$_{HH}$ = 6.9 Hz, 1 H, –CH(CH₃)₂), 3.28 (sept, ³J$_{HH}$ = 6.6 Hz, 1 H, –CH(CH₃)₂), 3.14 (sept, ³J$_{HH}$ = 6.9 Hz, 1 H, –CH(CH₃)₂), 2.73 (s, 1 H, SbH), 1.69 (s, 15 H, Cp*), 1.65, 1.63 (m, 9 H, ArNCCH₃, –CH(CH₃)₂), 1.56 (d, ³J$_{HH}$ = 6.6 Hz, 3 H, –CH(CH₃)₂), 1.34

(dd, ³J$_{HH}$ = 6.6 Hz, 6 H, –CH(CH₃)₂), 1.23 (d, ³J$_{HH}$ = 6.9 Hz, 3 H, –CH(CH₃)₂), 1.21 (d, ³J$_{HH}$ = 6.9 Hz, 3 H, –CH(CH₃)₂), 1.00 (d, ³J$_{HH}$ = 6.6 Hz, 3 H, –CH(CH₃)₂), 0.99 (d, ³J$_{HH}$ = 6.6 Hz, 3 H, –CH(CH₃)₂). ¹³C NMR (C₆D₆, 75.5 MHz): δ 168.85, 168.81 (ArNCCH₃), 146.62, 146.26, 142.84, 142.65, 141.45, 141.41, 125.76, 125.47, 124.12 (C₆H₃), 123.98 (C₅Me₅), 97.76 (γ-CH), 30.06, 29.87, 28.19, 28.10 (–CH (CH₃)₂), 26.95, 26.70, 25.02, 24.87, 24.81, 24.72, 24.31, 24.13 (–CH(CH₃)₂), 23.94, 23.76 (ArNCCH₃), 13.64 (C₅Me₅).

**Synthesis of L(Cl)GaSb(Cl)Cp* 3.** A mixture of LGa (0.1 g, 0.205 mmol) and Cp*SbCl₂ (0.067 g, 0.205 mmol) in 1 mL of benzene was stirred at room temperature for 1 h. The solvents were removed under reduced pressure and the residue was dissolved in 2 mL of hexane. The solution was stored at room temperature for 3 h to yield yellow crystals of **3**. Yield: 0.11 g (0.135 mmol, 66%). M.p. 130 °C (dec.). Anal. Calcd. for C₃₉H₅₆N₂Cl₂GaSb: C, 57.46; H, 6.92; N, 3.44. Found: C, 57.50; H, 6.98; N, 3.39%. IR (neat): ν 2960, 2913, 2855, 1546, 1528, 1435, 1388, 1312, 1260, 1179, 1097, 1016, 934, 864, 794, 759, 642, 583, 525, 443, 414 cm⁻¹. ¹H NMR (C₆D₆, 300.1 MHz): δ 7.26–7.02 (m, 6 H, C₆H₃(iPr)₂), 4.86 (s, 1 H, γ-CH), 3.86 (sept, ³J$_{HH}$ = 6.9 Hz, 1 H, –CH(CH₃)₂), 3.54 (sept, ³J$_{HH}$ = 6.6 Hz, 1 H, –CH (CH₃)₂), 3.45 (sept, ³J$_{HH}$ = 6.9 Hz, 1 H, –CH(CH₃)₂), 3.25 (sept, ³J$_{HH}$ = 6.9 Hz, 1 H, –CH(CH₃)₂), 1.76 (d, ³J$_{HH}$ = 6.6 Hz, 3 H, –CH(CH₃)₂), 1.73 (s, 15 H, Cp*), 1.57 (s, 3 H, ArNCCH₃), 1.53 (s, 3 H, ArNCCH₃), 1.53 (d, ³J$_{HH}$ = 6.6 Hz, 3 H, –CH(CH₃)₂), 1.38 (d, ³J$_{HH}$ = 6.9 Hz, 3 H, –CH(CH₃)₂), 1.27 (dd, ³J$_{HH}$ = 6.6 Hz, 6 H, –CH (CH₃)₂), 1.19 (d, ³J$_{HH}$ = 6.6 Hz, 3 H, –CH(CH₃)₂), 1.03 (d, ³J$_{HH}$ = 6.6 Hz, 3 H, –CH (CH₃)₂), 0.94 (d, ³J$_{HH}$ = 6.9 Hz, 3 H, –CH(CH₃)₂). ¹³C NMR (C₆D₆, 75.5 MHz): δ 169.96, 168.87 (ArNCCH₃), 146.77, 145.68, 142.45, 142.17, 141.98, 126.35, 125.79, 124.08 (C₆H₃), 122.92 (C₅Me₅), 97.71 (γ-CH), 30.89, 29.99, 28.32, 27.95 (–CH (CH₃)₂), 26.38, 26.28, 25.31, 24.91, 24.73, 24.40, 24.23 (–CH(CH₃)₂), 23.86, 23.58 (ArNCCH₃), 11.88 (C₅Me₅).

**Synthesis of [L(I)Ga]$_2$Bi· 4.** Volume of 5 mL of toluene was added to a mixture of LGa (0.30 g, 0.62 mmol) and Cp*BiI$_2$ (0.184 g, 0.31 mmol) at −40 °C. The dark-red suspension was slowly warmed to room temperature over a period of 4 h to give a clear red solution. All volatiles were removed under reduced pressure and the residue was washed three times with 5 mL of *n*-hexane to give **4** as a red micro-crystalline powder. Yield: 316 mg (0.22 mmol, 71%). M.p. 100 °C (dec.). Anal. Calcd. for C$_{65}$H$_{90}$BiGa$_2$I$_2$N$_4$ (**4**·toluene): C, 51.04; H, 5.93; N, 3.67. Found: C, 51.8; H, 5.99; N, 3.84%. IR (neat): ν 3060, 3021, 2961, 2922, 2864, 1523, 1437, 1380, 1314, 1258, 1174, 1099, 1018, 935, 859, 796, 757, 727, 693, 635, 525, 463, 438 cm$^{-1}$.

**Synthesis of LGaSbGa(Cl)L 5.** A mixture of **1** (0.120 g, 0.102 mmol) and KC$_8$ (0.014 g, 0.102 mmol) in 1 mL of benzene was stirred at room temperature for 1 h. The reaction mixture was filtered and the precipitate was washed with 1 mL of benzene. The solvent from the combined filtrates was removed under reduced pressure and the residue was dissolved in 1 mL of hexane. The dark-red solution was kept at room temperature for 2 days to afford red crystals of **5**. Yield: 70 mg (0.062 mmol, 60%). M.p. 226 °C. Anal. Calcd. for C$_{58}$H$_{82}$N$_4$ClGa$_2$Sb: C, 61.54; H, 7.30; N, 4.95. Found: C, 61.65; H, 7.22; N, 4.91%. IR (neat): ν 2957, 2925, 2863, 1523, 1436, 1380, 1311, 1255, 1174, 1106, 1019, 938, 856, 794, 757, 632, 526, 495, 445 cm$^{-1}$. $^{1}$H NMR (toluene-$d_8$, 500 MHz): δ 7.16–7.06 (m, 6 H, C$_6$H$_3$(iPr)$_2$), 4.87 (s, 1 H, γ-CH), 3.15 (sept, 4 H, −CH(CH$_3$)$_2$), 1.51 (s, 6 H, ArNCCH$_3$), 1.24 (d, 12 H, $^{3}J_{HH}$ = 7.0 Hz, -CH(CH$_3$)$_2$), 1.10 (d, 12 H, $^{3}J_{HH}$ = 6.5 Hz, −CH(CH$_3$)$_2$). $^{13}$C NMR (toluene-$d_8$, 150 MHz): δ 168.09 (ArNCCH$_3$), 143.71, 141.60, 127.27, 124.31 (C$_6$H$_3$), 98.28 (γ-CH), 29.01 (−CH(CH$_3$)$_2$), 26.32 (-CH(CH$_3$)$_2$), 24.40 (−CH (CH$_3$)$_2$), 24.30 (ArNCCH$_3$).

**Synthesis of LGaClF.** A mixture of **1** (28 mg, 0.024 mmol) and NOBF$_4$ (5.6 mg, 0.048 mmol) in 0.5 mL of benzene was sonicated at room temperature for 1 h. The solvents were removed under reduced pressure and the resulting residue was extracted with *n*-hexane (2 × 5 mL). The extract solution was concentrated to 1 mL and stored at room temperature for 1 day to afford colorless crystals of LGaClF. Yield: 15.4 mg (0.028 mmol, 59 %). Anal. Calcd. for C$_{29}$H$_{41}$N$_2$ClFGa: C, 64.29; H, 7.63; N, 5.17. Found: C, 64.22; H, 7.59; N, 5.21%. IR (neat): ν 3058, 2959, 2926, 2867, 1552, 1524, 1460, 1442, 1383, 1317, 1259, 1179, 1105, 1055, 1022, 946, 935, 878, 806, 796, 765, 756, 727, 716, 636, 608, 533, 443, 403 cm$^{-1}$. $^{1}$H NMR (C$_6$D$_6$, 300 MHz): δ 7.18–7.05 (m, 6 H, C$_6$H$_3$(iPr)$_2$), 4.80 (s, 1 H, γ-CH), 3.57 (sept, $^{3}J_{HH}$ = 6.9 Hz, 2 H, −CH(CH$_3$)$_2$), 3.19 (sept, $^{3}J_{HH}$ = 6.9 Hz, 2 H, −CH(CH$_3$)$_2$), 1.52 (s, 6 H, ArNCCH$_3$), 1.46 (d, $^{3}J_{HH}$ = 6.6 Hz, 6 H, −CH(CH$_3$)$_2$), 1.41 (d, $^{3}J_{HH}$ = 6.6 Hz, 6 H, −CH(CH$_3$)$_2$), 1.15 (d, $^{3}J_{HH}$ = 6.9 Hz, 6 H, −CH(CH$_3$)$_2$), 1.09 (d, $^{3}J_{HH}$ = 6.6 Hz, 6 H, −CH(CH$_3$)$_2$). $^{13}$C NMR (C$_6$D$_6$, 75.5 MHz): δ 171.91 (ArNCCH$_3$), 145.21, 143.77, 138.43, 128.91, 125.09, 124.48 (C$_6$H$_3$), 96.60 (γ-CH), 29.00 (−CH(CH$_3$)$_2$), 28.33 (−CH(CH$_3$)$_2$), 25.45 (−CH(CH$_3$)$_2$), 24.78 (−CH(CH$_3$)$_2$), 24.56 (−CH(CH$_3$)$_2$), 24.44 (−CH(CH$_3$)$_2$), 23.43 (ArNCCH$_3$). $^{19}$F NMR (C$_6$D$_6$, 282 MHz): δ −177.67.

**Data availability.** The crystallographic data have been deposited with the Cambridge Crystallographic Data Centre as supplementary publication nos. CCDC-1575886 (**1**), CCDC-1575134 (**2**), CCDC-1575887 (**3**), CCDC-1575135 (**4 · 2C$_7$H$_8$**), CCDC-1575885 (**5**), and CCDC-1580689 (**LGaClF**). Copies of the data can be obtained free of charge on application to CCDC, 12 Union Road, Cambridge, CB21EZ (fax: (+ 44) 1223/336033; e-mail: deposit@ccdc.cam-ak.uk). The authors declare that all other data supporting the findings of this study are available within the paper [and its supplementary information files].

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

## Acknowledgements

Financial support from the University of Duisburg-Essen and the German Research Foundation DFG (S.S.; research grant SCHU 1069/22-1), the Alexander von Humboldt Stiftung (G.E.C.) and the Max- Planck-Gesellschaft (G.E.C.) is gratefully acknowledged. We also thank E. Hammes (Heinrich Heine University Düsseldorf) for technical support and Peter G. Jones (TU Braunschweig, Germany) for the crystallographic data measurement, procession and the solution of **3**.

## Author contributions

C.G. and C.H. designed the study under the supervision of S.S., performed reactions and collected and analyzed the data. G.E.C. performed the EPR studies and theoretical calculations. E.B. collected the SQUID data. Crystallographic data were collected and refined by C.W. (**1**, **3**, **5**, **LGaClF**) and W.F. (**2**, **4** · 2C₇H₈). C.G., C.H., G.E.C. and S.S. wrote the paper. All the authors discussed the results and commented on the manuscript.

## Additional information

**Competing interests:** The authors declare no competing financial interests.

