## [Peer Review File · Nature Communications]

REVIEWERS' COMMENTS:

Reviewer #1 (Remarks to the Author):

This is a very nice piece of work, experimentally challenging and I thought most carefully and thoroughly carried out. The Sb and Bi radicals are generally very unstable species and their stabilisation by coordination to Ga residues is impressive. The Ga=S is also a significant result. Overall I thought this was an excellent paper, I have no criticisms or suggestions for additional work/measurements and I strongly recommend publication. I am sure it is a very significant contribution to the field and will be of much interest to a wide variety of main group coordination/organometallic chemists. Whilst for many years antimony and bismuth chemistry was considered "similar but less interesting than P chemistry", it is currently an area of much research activity and has areas which have no analogue in phosphorus chemistry- for example hypervalency of metal coordinated Sb or Bi groups (Benjamin and Reid, *Coord. Chem. Rev.* 297-298, 168, 2015). The very different behaviour of P versus Sb/Bi centred radicals and the general rarity of double bonds Sb/Bi=O/S etc, are now getting the effort they deserve and this paper is an important contribution.

Reviewer #2 (Remarks to the Author):

This is a very nice paper reporting several important findings. Among them, it is demonstrated for the first time that the coordination of an electropositive metal at the radical center has a stabilizing effect. This reviewer is surprised to read that the spin density is mainly localized at the meta center, which is in marked contrast to previous radicals which were stabilized by a CAAC (for example in gold chemistry, *Angew. Chem. Int. Ed.* 2013, 52, 8964). The isolation of a compound featuring a Ga=Sb double bond is also a highlight, although the p-bonding contribution is only 9.6 kcal/mol.

A very minor comment: The X-ray discussion is too long. For example, the group of symmetry is not really of interest in the main text. I believe that other very technical descriptions could be shortened since it buries the important findings.

Response to Referee Comments

Reviewer 1

This is a very nice piece of work, experimentally challenging and I thought most carefully and thoroughly carried out. The Sb and Bi radicals are generally very unstable species and their stabilisation by coordination to Ga residues is impressive. The Ga=S is also a significant result. Overall I thought this was an excellent paper, I have no criticisms or suggestions for additional work/measurements and I strongly recommend publication. I am sure it is a very significant contribution to the field and will be of much interest to a wide variety of main group coordination/organometallic chemists. Whilst for many years antimony and bismuth chemistry was considered "similar but less interesting than P chemistry", it is currently an area of much research activity and has areas which have no analogue in phosphorus chemistry- for example hypervalency of metal coordinated Sb or Bi groups (Benjamin and Reid, *Coord. Chem. Rev.* 297-298, 168, 2015). The very different behaviour of P versus Sb/Bi centred radicals and the general rarity of double bonds Sb/Bi=O/S etc, are now getting the effort they deserve and this paper is an important contribution.

Answer by the authors: We are thankful to reviewer 1 for recommending the manuscript for publication in Nature Communication. No changes were requested.

Reviewer 2:

This is a very nice paper reporting several important findings. Among them, it is demonstrated for the first time that the coordination of an electropositive metal at the radical center has a stabilizing effect. This reviewer is surprised to read that the spin density is mainly localized at the meta center, which is in marked contrast to previous radicals which were stabilized by a CAAC (for example in gold chemistry, *Angew. Chem. Int. Ed.* 2013, 52, 8964). The isolation of a compound featuring a Ga=Sb double bond is also a highlight, although the p-bonding contribution is only 9.6 kcal/mol.

A very minor comment: The X-ray discussion is too long. For example, the group of symmetry is not really of interest in the main text. I believe that other very technical descriptions could be shortened since it buries the important findings.

Answer by the authors: We are thankful to reviewer 1 for recommending the manuscript for publication in Nature Communication. As suggested, the X-ray discussion was shortened.

Additional remark on the A alert in the checkcif report for CCDC_1575134

This alert is caused by H1, which is only partially constrained. Apparently checkcif considers it to be refined freely despite the constraint.